# Simultaneous Down-Regulation of Intracellular MicroRNA-21 and hTERT mRNA Using AS1411-Functionallized Gold Nanoprobes to Achieve Targeted Anti-Tumor Therapy

**DOI:** 10.3390/nano14231956

**Published:** 2024-12-05

**Authors:** Qinghong Ji, Qiangqiang Yang, Mengyao Ou, Min Hong

**Affiliations:** School of Chemistry and Chemical Engineering, Liaocheng University, Liaocheng 252059, China; 17861826032@163.com (Q.J.); mortalyang@163.com (Q.Y.); 15275820795@163.com (M.O.)

**Keywords:** gene regulation, microRNA-21, hTERT mRNA, gold nanoparticles, anti-tumor

## Abstract

Telomerase presents over-expression in most cancer cells and has been used as a near-universal marker of cancer. Studies have revealed that inhibiting telomerase activity by utilizing oligonucleotides to down-regulate the expression of intracellular human telomerase reverse-transcriptase (hTERT) mRNA is an effective method of achieving anti-tumor therapy. Considering that oncogenic microRNA-21 has been proven to indirectly up-regulate hTERT expression and drive cancer metastasis and aggression through increased telomerase activity, here, we constructed an AS1411-functionallized oligonucleotide-conjugated gold nanoprobe (Au nanoprobe) to simultaneously down-regulate intracellular microRNA-21 and hTERT mRNA by using anti-sense oligonucleotide technology to explore their targeted anti-tumor therapy effect. In vitro cell studies demonstrated that Au nanoprobes could effectively induce apoptosis and inhibit the proliferation of cancer cells by down-regulating intracellular hTERT activity. In vivo imaging and anti-tumor studies revealed that Au nanoprobes could accumulate at the tumor site and inhibit the growth of MCF-7 tumor xenografted on balb/c nude mice, thus having potential for anti-tumor therapy.

## 1. Introduction

Considering the high telomerase activity in most cancer cells, cancer treatment specifically targeting telomerase has attracted people’s interest [1,2]. As one of the main subunits of telomerase, human telomerase reverse-transcriptase catalytic subunit (hTERT) has been proved to be the rate-limiting component of telomerase activity and has thus become an important target for telomerase regulation [3,4]. Some studies have pointed out that the regulation of hTERT mRNA by anti-sense technology can effectively induce apoptosis of cancer cells [5], and the drug Imetelstat (GRN163L) based on this principle has long been in clinical trials [6].

Many studies have shown that a variety of microRNAs in cancer cells, such as carcinogenic microRNAs including microRNA-19b, microRNA-346 and microRNA-21, could directly or indirectly up-regulate the expression of hTERT so as to promote cancer invasiveness, oxidative stress, genomic instability, and cell proliferation, as well as evasion of apoptosis [7,8,9]. Among them, microRNA-21 has been found to be indirectly associated with hTERT up-regulation in colorectal cancer and malignant melanoma cells by down-regulating PTEN (a tumor-suppressor gene) and then activating the PI3K/Akt pathway [10]. Also, microRNA-21 was proven to enhance carcinogenesis through STAT3, and reduced expression of both hTERT and STAT3 as well as slowed tumor growth were observed when microRNA-21 was knocked down in murine glioblastoma xenografts [11]. Moreover, if the inhibition of carcinogenic microRNAs is combined with telomerase therapy, the resistance effect in telomerase therapy can be reduced [12]. Therefore, the down-regulation of carcinogenic microRNAs and hTERT mRNA by anti-sense oligonucleotide technology will play a positive role in combating cancer. However, the nucleic acid fragment does not easily enter the cell directly; appropriate carriers are needed to improve the intake rate of anti-sense oligonucleotides.

The structure of tumor vessels is abnormal, with wide interendothelial junctions and a large number of fenestrations. Therefore, after intravenous administration, nanoparticles easily extravasate through leaky vasculatures and accumulate in tumors [13]. In addition, targeting ligands modified on the nanoparticle surface can specifically recognize tumors and bind to over-expressed receptors with a high affinity in the target region, which would induce nanomedicines extravasating into tumors through active trans-endothelial mechanisms [14]. Among the nanoparticle-based drug delivery systems that have been developed, gold nanoparticles (AuNPs) have become the preferred carrier for researchers [15,16] due to their several advantages of good biosecurity, stability, surface functionalization and fluorescence-quenching property. Also, AuNPs have been proved to penetrate the tumor vascular system and improve tumoral vascular leakiness and increase the tumoral accessibility of anti-tumor therapeutics, subsequently enhancing therapeutic efficiency [17]. Altogether, targeting ligand-modified AuNP-based drug delivery systems presents great application potential in the field of targeted anti-tumor therapy.

To expand the application of AuNPs in tumor theranostics, previously, we designed two AuNP-based nucleic acid probes loaded with anti-sense oligonucleotide sequences to achieve in situ detection, fluorescence imaging and down-regulation of intracellular hTERT mRNA or microRNA-21 [18,19]. While the effectiveness of AuNP-based nucleic acid probes in inducing cancer cell apoptosis by anti-sense technology has been verified at the cell level, in vivo studies are needed to show their anti-tumor therapy potential. As a continuation of this series of research work, here, we designed a gold nanoprobe (Au nanoprobe) containing two types of anti-sense sequences, in which one type of anti-sense sequence was used for down-regulating hTERT mRNA and the other was designed for silencing microRNA-21. In addition, in order to achieve targeted delivery to tumors and improve cancer-cell uptake of nanomedicines, the aptamer AS1411 that recognizes cancer cells through high affinity with the surface over-expressed nucleolin [20] was modified on the surface of Au nanoprobes with disulfide bonds as linkers. The in vitro and in vivo anti-cancer effect of Au nanoprobes was investigated here.

## 2. Materials and Methods

### 2.1. Preparing Au NPs and Au Nanoprobes

AuNPs were synthesized according to the literature previously reported [21]. Firstly, 50 mL of HAuCl_4_ (1 mM) solution was heated to 100 °C. Next, 10 mL of trisodium citrate solution (38.8 mM) was quickly added the above boiled HAuCl_4_ solution, and the mixed solution was stirred thoroughly at 100 °C for 15 min to obtain the dark red AuNP solution. The size and morphology of AuNPs were characterized by the JEM-2100 transmission electron microscope (JEOL, Tokyo, Japan).

Au nanoprobes are prepared according to the following procedure. All DNA sequences (1 OD) (Appendix A) were individually dissolved in 100 μL DEPC water. HS-anti-hTERT-DNA and Cy3-hTERT-DNA were mixed in the molar ratio of 1:1.2, and the mixture was heated to 75 °C and held for 10 min. Then, the mixture was naturally cooled to room temperature and incubated under dark conditions for 12 h to obtain the hybridized hTERT-related DNA duplexes (HS-anti-hTERT-DNA/Cy3-hTERT-DNA). With the same procedure, hybridized microRNA-21-related DNA duplexes (HS-miRNA-21-DNA/Cy5-AS1411-anti-miRNA-21-DNA) were also prepared. Next, the hybridized hTERT- and microRNA-21-related DNA duplex systems were mixed together and further reacted with 4 mL AuNP solution. The final mixtures were incubated with rotation at room temperature for 24 h and further inactivated with PBS solution (400 μL) thrice with an interval time of 10 h. Then, the reaction mixture was centrifuged and the precipitate was washed with PBS solution thrice to discard the unbound DNA sequences. Finally, the newly prepared Au nanoprobes were dispersed in PBS solution (4 mL) and stored at 4 °C for future studies. The ultraviolet visible (UV-Vis) absorption spectrum and Raman spectrum of Au nanoprobes were determined, and the concentration of Au nanoprobes was calculated by measuring their extinction at 524 nm (*ε* = 2.7 × 10^8^ L·mol^−1^·cm^−1^).

### 2.2. Evaluating the Amount of DNA Duplexes Bound on Each Au Nanoprobe

The amount of HS-anti-hTERT-DNA/Cy3-hTERT-DNA or HS-miRNA-21-DNA/Cy5-AS1411-anti-miRNA-21 duplexes bound on each Au nanoprobe was evaluated according to the previously reported protocol [22]. Briefly, different concentrations of mercaptoethanol (0, 1, 1.5, 2, 3, 5, and 10 mM) were individually added to the probe solutions (1.5 nM). After incubation overnight with shaking at room temperature, DNA duplexes were gradually released by the competitive binding of mercaptoethanol with AuNPs. Then, the released DNA duplexes were separated from AuNPs through centrifugation, and the fluorescence intensity of suspension was determined by using an F-7000 spectrofluorometer (Hitachi, Tokyo, Japan) with excitation wavelengths of 530 and 630 nm. A standard linear calibration curve was prepared with known concentrations of two types of DNA duplexes (0, 10, 20, 40, 60, 80, 100, 120, 150, and 200 nM) with identical buffer pH, ionic strength and mercaptoethanol concentrations. The amount of HS-anti-hTERT-DNA/Cy3-hTERT-DNA or HS-miRNA-21-DNA/Cy5-AS1411-anti-miRNA-21 duplexes bound on each Au nanoprobe was calculated by referring the fluorescence intensity of the supernatant containing DNA duplexes collected after the prepared probes were incubated with mercaptoethanol (10 mM) to the standard curve.

### 2.3. Determining the Fluorescence Response of Au Nanoprobes to Target DNA

In order to verify the responsiveness of Au nanoprobes to the hTERT mRNA or microRNA-21-related target DNA, a series of Target-hTERT-DNA or Target-miRNA-21-DNA (0, 100, 200, 600, and 1000 nM) were individually mixed with Au nanoprobes (200 μL, 1.5 nM). All mixtures were incubated at 37 °C for 4 h. Then, the fluorescence intensity of different system was determined using the F-7000 spectrofluorometer (Hitachi, Japan) with excitation wavelengths of 530 and 630 nm.

### 2.4. Determining the Fluorescence Response of Au Nanoprobes to Intracellular MicroRNA-21 and hTERT mRNA

Two telomerase-positive cancer cell lines, including HeLa (human cervical cancer cells) and MCF-7 (human breast cancer cells), were cultured in the DMEM medium (GIBCO) containing fetal bovine serum and Penicillin-Streptomycin (100 µg·mL^−1^) with a ratio of 9:1:0.1 at 37 °C in a humidified atmosphere containing 5% CO_2_. The cell number was determined using the Petroff–Hausser cell counter (USA, Horsham, PA, USA).

The fluorescence responsiveness of Au nanoprobes to intracellular microRNA-21 and hTERT mRNA was studied with the cell lysates and living cells, respectively. The experimental process was as follows.

(i)Cell lysate analysis: Cell lysates were obtained by breaking down MCF-7 cells (1 × 10^6^) using an ultrasonic disruptor. Au nanoprobes (1.5 nM) were incubated with the freshly prepared cell extracts at 37 °C for 4 h. The fluorescence intensity levels of the experimental systems were determined by using the F-7000 spectrofluorometer (Hitachi, Japan) with excitation wavelengths of 530 and 630 nm.(ii)In situ fluorescence imaging: MCF-7 or HeLa cells (0.4 mL, 1 × 10^6^ mL^−1^) were, respectively, seeded in a 20 mm glass-bottom confocal dish. After 24 h, Au nanoprobes (1.5 nM) were incubated with cells for 4 h. Then, cells were washed with PBS thrice and observed by LSM880 confocal laser scanning microscopy (CLSM, Zeiss, Jena, Germany). The fluorescence signals of Cy3 and Cy5 of Au nanoprobes responsive to hTERT mRNA and microRNA-21 were excited with wavelengths of 543 and 633 nm, respectively.

### 2.5. Quantifying the Uptake of Au Nanoprobes in Cancer Cells 

MCF-7 or HeLa cells (1 × 10^5^ mL^−1^) were seeded in DMEM with 10% fetal bovine serum. After 24 h, cells were incubated with the Au nanoprobes or Control–Au nanoprobes (1.5 nM) for 3 h. At the end of the incubation period, the cells were washed three times with PBS buffer and trypsinized to remove them from the bottom. Then, the cells were counted and collected in centrifuge tubes. Next, cells were fully digested in 0.2 mL of aqua regia (3:1 hydrochloric acid/nitric acid) and sonicated for 30 min. Following incubation overnight, the sample was diluted to 3 mL using ultrapure water. The sample was then analyzed for total gold content by inductively coupled plasma–atomic emission spectroscopy (ICP-AES), and the measurement was repeated three times. The amount of Au as determined by ICP-AES analysis was then converted to the number of nanoparticles using the average nanoparticle diameter, as determined via TEM imaging. 

### 2.6. Analyzing the Intracellular hTERT mRNA and microRNA-21 Level

HeLa and MCF-7 cells (5 × 10^6^) in the logarithmic growth phase were treated with AuNP probes (1.5 nM) or HS–control–DNA/Control–DNA duplex-functionalized probes (Control––Au nanoprobes, 1.5 nM) for different times (12, 24, 48, or 72 h). Then, total RNA from the tested cells was extracted using Trizol total RNA isolation reagent (TIANGEN, Beijing, China) according to the manufacturer’s instructions. The cDNA was reverse- transcribed using a QuantiNova Reverse Transcription Kit (Qiagen, Duesseldorf, Germany). The reactions were incubated in a thermal cycler for 60 min at 37 °C, 5 min at 95 °C, and then held at 4 °C. Real-time quantitative, reverse-transcription polymerase chain reaction (qRT-PCR) was performed using the QuantStudio™ 5 Real-Time PCR system (Applied Biosystems, Waltham, MA, USA) with specific microRNA-21 primers from a commercial kit (miScript Primer Assays, Qiagen) and miScript SYBR^®^ Green PCR Kit (Qiagen, Hilden, Germany). Relative level of microRNA-21 was calculated from the quantity of microRNA-21 PCR products and the quantity of RNU6B PCR products and normalized to the expression level in untreated cells using the 2^−ΔΔ*CT*^ method [ΔΔ*CT* = (*CT*_miRNA-21_ − *CT*_U6-RNA_)_experimental group_ − (*CT*_miRNA-21_ − *CT*_U6-RNA_)_untreated group_]. The reaction proceeded as follows: 1 cycle of 95 °C for 15 min, followed by 40 cycles of 94 °C for 15 s, 55 °C for 30 s, and 70 °C for 30 s.

With a similar procedure, the relative level of hTERT mRNA was also determined using qRT-PCR. The sequences of forward and reverse primers of hTERT and GAPDH are given in Appendix A. The reaction proceeded as follows: 1 cycle of 50 °C for 2 min and 1 cycle of 95 °C for 2 min were followed by 40 cycles of 95 °C for 15 s, 55 °C for 15 s, and 72 °C for 1 min.

### 2.7. Analyzing the Intracellular hTERT Activity

MCF-7 cells (5 × 10^5^ cells/well) were inoculated in 6-well plates and cultured for 24 h. Then, all cell samples were divided into three groups and were incubated with PBS, Control––Au nanoprobes (1.5 nM), or Au nanoprobes (1.5 nM), respectively, for 48 or 72 h. After the treatment, hTERT in different cell samples was extracted according to the following procedure. Firstly, 1 × 10^6^ cells were dispensed in a 1.5 mL EP tube, washed thrice with ice-cold PBS (0.1 M, pH 7.4) through centrifugation, and resuspended in ice-cold CHAPS lysis buffer (200 µL) containing 10 mM Tris-HCl, pH 7.5, 1 mM MgCl_2_, 1 mM EGTA, 0.1 mM PMSF, 0.5% CHAPS and 10% glycerol. The mixture was incubated for 30 min on ice and centrifuged at 16,000 rpm at 4 °C for 20 min. The supernatant was collected as cell extract for analysis. To quantify the hTERT activity in different samples, a standard curve was constructed using a commercial hTERT activity ELISA Kit (Shanghai Kepeirui Biotech. Co. Ltd., Shanghai, China). The hTERT activity level in different cell extracts was determined according to the procedure given by the ELISA Kit and referring to the standard curve.

### 2.8. Determining the Pro-Apoptosis Effect and In Vitro Cytotoxicities of Au Nanoprobes

The pro-apoptosis effect of Au nanoprobes on MCF-7 cells was investigated using the AnnexinV-FITC/PI method. MCF-7 (5 × 10^5^ cells/well) cells were inoculated in 6-well plates. Cells were incubated with PBS (blank group) or Au nanoprobes (1.5 nM), respectively, for 24, 48, or 72 h. Then, cells were stained according to the procedure given by the commercial Annexin V-FITC/PI apoptosis kit (Beyotime, Beijing, China) and collected after trypsinization treatment for the cell apoptosis analysis determined by Guava easyCyte 5HT flow cytometer (Millipore, Waltham, MA, USA). The apoptosis data were analyzed by FlowJo v10 software.

The in vitro cytotoxicities of Au nanoprobes against HeLa and MCF-7 cells were determined using the MTT method. Firstly, a certain number of cells were inoculated in 96-well plates (1 × 10^5^ cells/well). Then, 24 h later, cells were treated with Au nanoprobes (1.5 or 2 nM) for 24, 36, 48, or 72 h. For the blank or control groups, cells were treated with PBS or Control–Au nanoprobes (1.5 or 2 nM), respectively. Then, the cell medium was removed and replaced with 100 μL fresh medium containing 2.5 mg/mL of MTT. After 4 h, 100 μL DMSO was added to dissolve the formazan crystals after the removal of MTT solution. The absorbance at the wavelength of 490 nm was measured with a microplate reader. Cell survival was calculated by subtracting the optical density (OD) value of each well by that of the blank group.

### 2.9. In Vivo and Ex Vivo Fluorescence Imaging and Quantifying the Accumulation of Au Nanoprobes in Tumors

Xenograft tumor models of MCF-7 were built by subcutaneously injecting MCF-7 cells (1 × 10^6^) in 200 µL Matrigel into the right flank of female balb/c nude mice (3~4 weeks old). MCF-7-tumor-bearing balb/c nude mice with tumor volumes of ~300 mm^3^ were randomly divided into three groups (*n* = 3) and fasted 12 h before the experiment with free access to water. Then, biodistribution of Au nanoprobes was investigated after a single intravenous injection at a dose of 50 μL (6.5 nM). As controls, the other two groups of mice were individually administered with PBS or Control–Au nanoprobes (50 μL, 6.5 nM). At timed intervals, the mice were anesthetized and then imaged by using the PerkinElmer (Waltham, MA, USA) IVIS Spectrum In Vivo Imaging System for tracking the Cy5 modified anti-miRNA-21-DNA (excitation: 640 nm; emission: 680 nm; epi-illumination). In addition, at 60 min, two representative mice that were treated individually with PBS or Au nanoprobes were sacrificed by cervical dislocation and the tumors as well as main organs (hearts, livers, spleens, lungs, kidneys) were excised and imaged by the IVIS Spectrum system.

Finally, the tumor tissues were digested in concentrated nitric acid using a microwave digestion instrument, and the concentration of Au in Au nanoprobes or Control––Au nanoprobe-treated tumors was determined by ICP-AES. 

### 2.10. In Vivo Anti-Tumor Study

When the tumors reached approximately 100 mm^3^ (set as Day 0), MCF-7-tumor-bearing balb/c nude mice were randomly divided into three groups (n = 5): PBS blank group, Au nanoprobe group, and Control––Au nanoprobe group. Next, at different time intervals (Day 0, 2, 4, 6, 8, 10, and 12), mice were injected via tail vein with 50 µL different systems: PBS, Au nanoprobes (3 nM), and Control––Au nanoprobes (3 nM). The tumor volumes and the mice body weights were determined synchronously every two days. On day 28, all mice were sacrificed by cervical dislocation. All experiments were carried out in accordance with the National Guide for Care and Use of Laboratory Animals.

### 2.11. Statistical Analysis

All experiments were performed in triplicate, and all data are presented as mean and standard deviation. Data were analyzed using IBM SPSS Statistics 25. Values of *p* < 0.05 and *p* < 0.01 were considered statistically significant.

## 3. Results and Discussion

### 3.1. Design Mechanism of Au Nanoprobes

The Au nanoprobes are constructed by combining two types of DNA duplexes on AuNPs through the Au-S bonds (Figure 1). Among the two types of DNA duplexes, the HS-anti-hTERT-DNA/Cy3-hTERT-DNA duplex is composed of a thiol group modified DNA (HS-anti-hTERT-DNA) in which the part fragment at the 5′ end is designed as the hTERT mRNA anti-sense sequence and a Cy3 labeled DNA (Cy3-hTERT-DNA) that has the same sequence with part fragment of hTERT mRNA [23]. In the other DNA duplexes, HS-miRNA-21-DNA/Cy5-AS1411-anti-miRNA-21-DNA, the thiol-group-modified DNA (HS-miRNA-21-DNA) is designed having the same sequence with part fragment of microRNA-21 [24]. In Cy5-AS1411-anti-miRNA-21-DNA, the Cy5 is labeled at the 3′ end of microRNA-21 anti-sense sequence and the AS1411 aptamer is linked with the anti-sense sequence through a disulfide bond. When Au nanoprobes reach the tumor tissue, the high affinity and specificity of AS1411 with nucleolin over-expressed on the cancer cell membrane [25,26] will promote the endocytosis of the probes by cancer cells and enhance the uptake of Au nanoprobes in tumor tissue. In addition to the special optical property of AuNPs, the fluorescence signals of Cy3 and Cy5 can be quenched in the Au nanoprobes [18,19]. Upon encountering target hTERT mRNA and microRNA-21, Cy3-hTERT-DNA and Cy5-AS1411-anti-miRNA-21-DNA will dissociate from Au nanoprobes because of the competing hybridization between HS-anti-hTERT-DNA and hTERT mRNA as well as Cy5-AS1411-anti-miRNA-21-DNA and microRNA-21 by forming more stable duplexes with longer hybridizing fragments. This will induce the recovery of the fluorescence signals of Cy3 and Cy5 and achieve the in situ monitoring and silencing of intracellular hTERT mRNA and microRNA-21. At the same time, under the action of intracellular glutathione (GSH), the disulfide bond in the Cy5-AS1411-anti-miRNA-21-DNA sequences will be broken. After separating with AS1411, the antisense sequence anti-miRNA-21-DNA can play a better role of silencing microRNA-21. Finally, under the synergism of tumor-targeting delivery and the down-regulation of the expression of intracellular hTERT mRNA and microRNA-21, Au nanoprobes can exert effective anti-tumor effects in vivo.

### 3.2. Characterization of AuNPs and Au Nanoprobes

The size (~13 nm) and the spherical morphology of AuNPs were characterized by transmission electron microscopy (TEM) (Appendix A). UV–Vis absorption spectra of both AuNPs and Au nanoprobes were determined, and the results show a characteristic peak of AuNPs at 520 nm in the two spectral curves (Appendix A). But for Au nanoprobes, the characteristic absorption peak of nucleic acid at 260 nm is observed. In addition, a surface-enhanced Raman spectrum (SERS) of Au nanoprobes was measured, and characteristic peaks assigned to Cy3-hTERT-DNA (1148, 1363, and 1469 cm^−1^) and Cy5-AS1411-anti-miRNA-21-DNA (556, 931, 1307, and 1596 cm^−1^) could be observed (Appendix A). Dynamic light scattering (DLS) was also used to verify the hydrodynamic changes after AuNPs were modified with the nucleic acid sequences, which increased from 13.3 ± 0.4 nm (Appendix A) to 43.0 ± 0.3 nm (Appendix A). All these results indicate that Au nanoprobes were successfully prepared. 

### 3.3. Evaluation of the Amount of DNA Duplexes Bound on Each Au Nanoprobe

Due to the competition reaction between mercaptoethanol and thiol-group-modified DNA [22], HS-anti-hTERT-DNA/Cy3-hTERT-DNA or HS-miRNA-21-DNA/Cy5-AS1411-anti-miRNA-21 duplexes bound on AuNPs can be displaced by mercaptoethanol and dissociate from Au nanoprobes. This will induce the fluorescence recovery of Cy3 and Cy5 in two types of DNA duplexes. The studies revealed that mercaptoethanol with a concentration of 10 mM is enough to replace all DNA sequences bound on AuNPs. Therefore, based on the fluorescence intensity (Figure 1A,B) of the supernatant containing DNA duplexes collected from the incubation solution of Au nanoprobes and mercaptoethanol (10 mM) and the standard curve of two types of DNA duplexes (Appendix A), each AuNP was estimated to carry 47 ± 8 HS-anti-hTERT-DNA/Cy3-hTERT-DNA duplexes and 26 ± 5 HS-miRNA-21-DNA/Cy5-AS1411-anti-miRNA-21 duplexes.

### 3.4. In Vitro Response of Au Nanoprobes to hTERT mRNA and microRNA-21-Related Target DNA

In order to play the role of gene silencing, partial fragments of HS-anti-hTERT-DNA and Cy5-AS1411-anti-miRNA-21-DNA in the Au nanoprobes are designed as the anti-sense sequences of hTERT mRNA and microRNA-21. Therefore, when Au nanoprobes were incubated with the two target DNA that have the same sequences as microRNA-21 or the key fragment of hTERT mRNA (5′-GGUCGAUUGUGAACAUGGA-3′) (4015 nucleotides; accession no. AF015950) [23], which can be used as a gene silence site, Cy3-hTERT-DNA and Cy5-AS1411-anti-miRNA-21-DNA dissociated from Au nanoprobes and induced the recovery of the fluorescence signals of Cy3 and Cy5. As shown in Figure 1C,D, with the increase in concentrations of two types of target DNA, the fluorescence intensity of Cy3 and Cy5 correspondingly increased. This indicates more Cy3-hTERT-DNA and Cy5-AS1411-anti-miRNA-21-DNA were released from the Au nanoprobes due to the competition hybridization between HS-anti-hTERT-DNA and Target-hTERT-DNA as well as Cy5-AS1411-anti-miRNA-21-DNA and Target-miRNA-21-DNA. In addition, the fluorescence intensity of Cy3 (Figure 1E) and Cy5 (Figure 1F) was also significantly increased after the addition of cell extract (curve b). These results demonstrate the successful response of the Au nanoprobe to intracellular hTERT mRNA and microRNA-21.

### 3.5. In Situ Fluorescence Imaging of Intracellular hTERT mRNA and microRNA-21 Using Au Nanoprobes and Evaluating the Intracellular Amount of Au Nanoprobes

After validating the response of Au nanoprobes to two DNA or RNA targets in PBS solution or cell lysate, next, by using CLSM, we investigated the in situ fluorescence recovery of Au nanoprobes in two cancer cell lines, HeLa and MCF-7, which highly express hTERT mRNA and microRNA-21 [18,19,27]. As shown in Figure 2A, after incubation with Au nanoprobes for 4 h, fluorescence signals of both Cy3 (green) and Cy5 (red) under the excitation of 543 and 633 nm could be observed in two cell lines. This reveals that the hybridization between target RNA (hTERT mRNA and microRNA-21) and Au nanoprobes happened in two types of cancer cells.

Next, the uptake amounts of Au nanoprobes in two cancer cell lines were determined using ICP-AES. The studies revealed that the numbers of Au nanoprobes a cell can uptake were calculated to be 5.3 ± 0.4 × 10^4^ particles and 4.8 ± 0.6 × 10^4^ particles in HeLa and MCF-7 cells, respectively. With the same procedure, the uptake amounts of Control––Au nanoprobes in two cancer cell lines were also determined, which are 2.8 ± 0.3 × 10^4^ and 2.3 ± 0.6 × 10^4^ particles in HeLa and MCF-7 cells, respectively. By comparison, it can be concluded that AS1411 modified on Au nanoprobes enhanced the uptake of Au nanoprobes by the specific affinity with nucleolins that are over-expressed on the surface of HeLa and MCF-7 cells.

### 3.6. Analysis of the Expression Level of hTERT mRNA and microRNA-21 as Well as hTERT Activity in Cancer Cells Treated with Au Nanoprobes

As mentioned above, anti-hTERT-DNA and anti-miRNA-21-DNA in Au nanoprobes were designed as anti-sense sequences to reduce the expression of hTERT mRNA and microRNA-21 in cancer cells by using anti-sense oligonucleotide technology. After verifying the responsiveness of Au nanoprobes to intracellular hTERT mRNA and microRNA-21, we further evaluated their gene silence efficiency by the qRT-PCR technique. The PCR results (Figure 2B,C) indicated that both hTERT mRNA and microRNA-21 were down-regulated in the Au nanoprobe-treated HeLa and MCF-7 cells, especially for the samples that were treated with the Au nanoprobes for 72 h. In contrast, there is no obvious down-regulation for two RNA targets when these two types of cancer cells were treated with the Control––Au nanoprobes. This demonstrates that the Au nanoprobes designed here can effectively down-regulate the expression of intracellular hTERT mRNA and microRNA-21 simultaneously in cancer cells through gene silencing technology. Consistent with this, our studies also revealed that the repression of hTERT mRNA and microRNA-21 could significantly inhibit hTERT expression in MCF-7 cells (Figure 2D), which has been proved to be caused by the regulation of STATS or PTEN expression [11,28].

### 3.7. Studies of the Pro-Apoptosis Effect and In Vitro Cytotoxicity of Au Nanoprobes

As hTERT expression is closely associated with telomerase activity, which is critical to the growth of cancer cells; the down-regulation of hTERT activity would induce apoptosis and inhibit the proliferation of cancer cells. Previously, we have investigated the pro-apoptosis effect of AuNP-based nucleic acid probes through the down-regulation of microRNA-21 or hTERT mRNA individually [18,19] by using anti-sense oligonucleotide technology. However, when relying solely on the single regulation of microRNA-21 or hTERT mRNA, its ability to inhibit cell proliferation is weak. The experimental results showed that after down-regulating hTERT mRNA in HeLa cells with AuNP-based nucleic acid probes for 72 h, the cell proliferation inhibition rate was only 28% [18]. Similarly, when only down-regulating microRNA-21 in MCF-7 cells, the inhibition rate of cell proliferation was just 20% [19]. To further deepen the research system, here, we investigated cellular apoptosis under the circumstances of simultaneously reducing the expression of microRNA-21 and hTERT mRNA. The flow cytometric analysis revealed that the apoptosis rate of cells treated by Au nanoprobes gradually increases with the increase in action time (34.54% at 24 h, 44.12% at 48 h, and 45.68% at 72 h) (Figure 3A,B). Also, this pro-apoptosis efficacy surpasses that caused by solely down-regulating microRNA-21 or hTERT mRNA with inhibitors (10~20%) [11,28].

Next, we further investigated the in vitro anticancer activity of Au nanoprobes with the MTT method. The studies indicate that Au nanoprobes significantly inhibited the proliferation of HeLa and MCF-7 cells in a concentration- and time-dependent manner. After 72 h of interaction with 2 nM Au nanoprobes, the proliferation inhibition rates of HeLa and MCF-7 cells reached 43% and 57% (Figure 3C,D), respectively, which were significantly higher than the cell proliferation inhibition rates achieved by solely down-regulating microRNA-21 or hTERT mRNA [18,19]. Comparatively, Control––Au nanoprobes did not present obvious in vitro cytotoxicities against HeLa and MCF-7 cells in the absence of effective gene regulation. Combining the above gene level and hTERT activity analysis, we can conclude that simultaneously decreased microRNA-21 and hTERT mRNA levels caused by Au nanoprobes through anti-sense oligonucleotide technology could lead to a more significant decrease in hTERT activity, thus more effectively inducing cancer cell apoptosis and exerting anti-proliferative effects.

### 3.8. In Vivo and Ex Vivo Imaging of Au Nanoprobes

Numerous studies have shown that the enhanced permeability and retention effect (EPR) related to nanostructures has a very limited impact on the targeted delivery of drugs to tumors. Therefore, researchers have turned to active targeted delivery by binding targeting ligands on the surface of nanomedicines. Among them, DNA aptamer-modified nanostructures as a tumor-targeting drug-delivery platform [29,30] have been widely reported. Among these studies, high binding affinity and specificity of the aptamer AS1411 with necleolin that is highly expressed on the surface of cancer cells have been widely studied to achieve targeted anti-tumor therapy [31,32]. In this work, to endow Au nanoprobes with tumor targeting, AS1411 with the sequence of 5′-d(GGT GGT GGT GGT TGT GGT GGT GGT GGA AAA AAA AA)-3′ was designed to link with anti-miRNA-21-DNA through a disulfide bond. This will realize tumor-targeted delivery and improve the cancer cell uptake rate of Au nanoprobes. Additionally, besides the passive and active tumor targeting delivery effect of AS1411-modified nanomedicines [33,34], AuNPs themselves have been proved to break through the tumor vascular basement membrane and cross the endothelial barrier through endothelial leakiness to improve nanotherapeutic delivery to tumors [35]. Therefore, the Au nanoprobes designed here should access the tumor to exert a therapeutic outcome. To verify this speculation, we next assessed the biodistribution of the Au nanoprobes after a single intravenous injection into MCF-7-tumor-bearing mice by utilizing in vivo fluorescence imaging. Considering the limit of weak penetration of Cy3 fluorescence, the fluorescence signal of Cy5 was used to perform in vivo imaging for tracking the response of Au nanoprobes to microRNA-21 in vivo and the biodistribution of Cy5-labeled anti-miRNA-21-DNA sequences. The results showed that after intravenous administration of Au nanoprobes in MCF-7-tumor-bearing mice, the Cy5 fluorescence signal was quickly observed at the abdomen and peritumoral sites within 10 min, and the intensity increased over a period of 40 min (Figure 4A,B). Also, the obvious Cy5 fluorescence appeared at the tumor sites in a period of 15~30 min. Furthermore, ex vivo fluorescence imaging of the main organs and tumors dissected from MCF-7 tumor-bearing mice after 60 min of administration showed bright fluorescence signal in the liver, kidney and tumor (Figure 4C). All these results demonstrate the effective accumulation of Au nanoprobes at the tumor site, and then the residues can be excreted through the liver and kidney.

After ex vivo fluorescence imaging, the accumulation of AuNPs in all tumor tissues were quantified by ICP-AES. The data demonstrate that after the injection of equivalent AuNP probes (50 μL, 6.5 nM) in the tail vein for 60 min, the amount of Au nanoprobes in the tumor site is 5.5-fold higher than that of Control–Au nanoprobes (Figure 4E), which proves that AS1411 could improve the tumor-targeted delivery and cancer cellular uptake of Au nanoprobes.

### 3.9. In Vivo Targeted Anti-Tumor Therapy

Encouraged by the above impressive in vitro pro-apoptosis and tumor-targeting results of Au nanoprobes, we next further investigated its in vivo anti-tumor therapeutic efficacy with an MCF-7 tumor model built by subcutaneously injecting MCF-7 cells into the right flank of balb/c nude mice. Au nanoprobes at a dosage of 3 nM were intravenously administered to mice every other day seven times. The tumor growth rate in terms of average tumor volume was recorded over a period of 29 days. The results showed that MCF-7 tumors in the PBS group grew gradually during the experiments. A similar phenomenon was also observed in the Control––Au nanoprobe-treated groups (Figure 5A,B). Although AuNPs have been reported to be able to induce innate immune responses through the process of immunogenic cell death [36], obviously, relying solely on this effect did not play an effective anti-tumor role here. Comparatively, the tumor volumes regressed gradually during the Au nanoprobe treatment, and this inhibition state of tumor proliferation could also remain in the following 10 days even after stopping administration. Afterwards, in the absence of the effect of Au nanoprobes, the tumor volume gradually increased. Thus, we can conclude that Au nanoprobes efficiently inhibited the MCF-7 tumor growth, and the simultaneous down-regulation of intratumoral microRNA-21 and hTERT mRNA caused by Au nanoprobes play a leading role in anti-tumor therapy.

During the experiments, the body weights of all mice were recorded, and the results showed that there was no significant difference for the average body weight of all three groups of mice (Figure 5C), indicating that the Au nanoprobes had no significant toxicity.

## 4. Conclusions

In conclusion, we constructed an AS1411 aptamer modified tumor-targeting Au nanoprobe by binding DNA duplexes on AuNPs with Au–S bonds. Under the synergism of passive and active targeting effects as well as the endothelial leakiness effect attributed to AuNPs, Au nanoprobes were able to accumulate at the tumor site. The response of Au nanoprobes to intracellular hTERT mRNA and microRNA-21 visually monitored by confocal fluorescence imaging caused the simultaneous down-regulation of the expression of hTERT mRNA and microRNA-21 by anti-sense oligonucleotide technology and induced cell apoptosis. The in vivo experiments demonstrate that Au nanoprobes could effectively inhibit tumor growth, showing great potential for the development of AuNP-based nanomedicines associated with telomerase therapy using gene regulation.

## Data Availability

The data that support the findings of this study are available on request from the corresponding author.

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
