# Peer review of "Simultaneous Down-Regulation of Intracellular MicroRNA-21 and hTERT mRNA Using AS1411-Functionallized Gold Nanoprobes to Achieve Targeted Anti-Tumor Therapy"

_nanomaterials, 2024, doi:10.3390/nano14231956_

Round 1
Reviewer 1 Report
Comments and Suggestions for Authors
The manuscript demonstrates the results of synthesis and research of functionalized gold nanoparticles with anti-sense oligonucleotide for targeted tumor therapy.
The manuscript is well structured, the methods and approaches used are modern and applicable. The text of the article is well written in high-quality academic language. The authors gave characteristics to nanoparticles with a complement
Meanwhile, in my opinion, some quantitative estimates of the identified effects are missing.
1. I would suggest doing FTIR or RAMAN spectroscopy analysis of the obtained nanoparticles to confirm their functionalization
2. I propose to quantify the accumulation of nanoparticles in cells (Figure 2). It is also important to assess exactly how they penetrate and where they are localized in the cell, as this determines the activity and biological effect in general.
3. I would recommend quantifying the fluorescence for Figure 4, given the availability of RAW data. The IVIS device allows you to do this.
4. It would be interesting to additionally demonstrate the quantitative accumulation of gold nanoparticles directly in tumor tissue by inductively coupled plasma mass spectrometry. In vivo fluorescence imaging does not quantify.
In general, I consider this work to be done very efficiently and of high scientific importance.
Author Response
Reviewer 1:
- In Figure, please label the graphs (a,b, etc.) with appropriate concentrations.
Response: We sincerely thank the reviewer’s suggestion. In Figure 1 E and F, we really forgot to provide the amount of cells for determining the fluorescence responsiveness of Au-nanoprobes to cell lysate. The corresponding data has been given in the caption of Figure 1.
- On page 7 line 296, please mention the units.
Response: We sincerely thank the reviewer’s comment. We have revised the description here.
- To show the targeting effect, The authors could use another control as an Au nanoprobe without AS1411.
Response: We sincerely thank the reviewer’s comment. As the reviewer mentioned, it is really necessary to show the targeting effect of AS1411 by using a control Au nanoprobe without AS1411. As shown in Figure 4A, in vivo fluorescence imaging were performed to assessed the biodistribution and tumor-targeting effect of the Au-nanoprobes after a single intravenous injection into MCF-7 tumor-bearing mice. Meanwhile, Control-Au-nanoprobes that were prepared with DNA duplexes but without AS1411 sequences were also used to prove the source of the fluorescence signal. We have determined the accumulation of gold nanoparticles in tumor tissue by inductively coupled plasma-atomic emission spectroscopy (ICP-AES) for these two groups of tumor samples. The ICP-AES data (Figure 4E) showed that the modification of AS1411 really enhanced the tumor-targeting of Au-nanoprobes.
- In Figure 3. the Au nanoprobe showed an apoptosis effect up to 72 hours, whereas in vivo studies, the Au nanoprobe residues, followed by the Cy5 fluorescence, decreased after 40 min. Please justify. Did the authors monitor the fluorescence after 24 hours or 48 hours?
Response: We sincerely thank the reviewer’s comment. In this work, telomerase activity in cancer cells was inhibited through gene regulation, thereby inducing cell apoptosis. By measuring the levels of microRNA-21 and hTERT mRNA in cancer cells treated with Au-nanoprobes, it can be concluded that gene regulation is a slow process. The results confirmed that the expression of regulated genes decreased sequentially from 24 hours to 48 hours, and then to 72 hours. Meanwhile, the activity of hTERT showed more significant decrease after treated for 72 hours compared to that of 48 hours. Therefore, the induction of apoptosis by Au-nanoprobes on cells should also be a slow process. However, after the Au-nanoprobes are injected into the mouse through the tail vein, it is quickly delivered to the tumor site along blood circulation, and enters cancer cells to interact with microRNA-21 and hTERT mRNA, then gradually undergoing metabolism. The tracking of the above process can be achieved through fluorescence signals. Due to the sensitivity limitation of in vivo imager, significant fluorescence signals can only be observed at live tumor sites when the signal is particularly strong. However, even if we cannot directly observe the fluorescence signal of the tumor site in vivo (Figure 4B), a very strong fluorescence signal can still be detected by peeling off the tumor (Figure 4C). This also proves the enrichment and reaction of Au-nanoprobes at the tumor site.
Reviewer 2:
The manuscript is well structured, the methods and approaches used are modern and applicable. The text of the article is well written in high-quality academic language. The authors gave characteristics to nanoparticles with a complement
Meanwhile, in my opinion, some quantitative estimates of the identified effects are missing.
- I would suggest doing FTIR or RAMAN spectroscopy analysis of the obtained nanoparticles to confirm their functionalization.
Response: We sincerely thank the reviewer’s suggestion. The Raman spectrum (Figure S2) has been determined and the corresponding data analysis has been added in the section “3.2. Characterization of AuNPs and Au-nanoprobes”.
- I propose to quantify the accumulation of nanoparticles in cells (Figure 2). It is also important to assess exactly how they penetrate and where they are localized in the cell, as this determines the activity and biological effect in general.
Response: We sincerely thank the reviewer’s suggestion. We have quantified the accumulation of Au nanoparticles in two cancer cell lines by using ICP-AES.
- I would recommend quantifying the fluorescence for Figure 4, given the availability of RAW data. The IVIS device allows you to do this.
Response: We sincerely thank the reviewer’s suggestion. The average fluorescence intensities shown in the Figure 4C analyzed by the IVIS imaging system have been provided in the Figure 4D.
- It would be interesting to additionally demonstrate the quantitative accumulation of gold nanoparticles directly in tumor tissue by inductively coupled plasma mass spectrometry. In vivo fluorescence imaging does not quantify.
Response: We sincerely thank the reviewer’s suggestion. We have quantified the accumulation of Au nanoparticles in tumor tissues by using ICP-AES.
Reviewer 2 Report
Comments and Suggestions for Authors
1. In Figure, please label the graphs (a,b, etc.) with appropriate concentrations.
2. On page 7 line 296, please mention the units.
3. To show the targeting effect, The authors could use another control as an Au nanoprobe without AS1411.
4. In Figure 3. the Au nanoprobe showed an apoptosis effect up to 72 hours, whereas in vivo studies, the Au nanoprobe residues, followed by the Cy5 fluorescence, decreased after 40 min. Please justify. Did the authors monitor the fluorescence after 24 hours or 48 hours?
Author Response

(The authors gave the same response as above.)
